# Metabolite Profiling of *Manilkara zapota* L. Leaves by High-Resolution Mass Spectrometry Coupled with ESI and APCI and In Vitro Antioxidant Activity, α-Glucosidase, and Elastase Inhibition Assays

**DOI:** 10.3390/ijms22010132

**Published:** 2020-12-24

**Authors:** Syful Islam, Md Badrul Alam, Hyeon-Jin Ann, Ji-Hyun Park, Sang-Han Lee, Sunghwan Kim

**Affiliations:** 1Department of Chemistry, Kyungpook National University, Daegu 41566, Korea; msi412@yahoo.com; 2Department of Environment, Munshiganj District Office, Munshiganj 1500, Bangladesh; 3Department of Food Science and Biotechnology, Kyungpook National University, Daegu 41566, Korea; mbalam@knu.ac.kr (M.B.A.); jiny345@knu.ac.kr (H.-J.A.); wlgus6744@knu.ac.kr (J.-H.P.); 4Inner Beauty/Antiaging Center, Food and Bio-Industry Research Institute, Kyungpook National University, Daegu 41566, Korea; 5knu BnC, Daegu 41566, Korea; 6Mass Spectrometry Converging Research Center and Green-Nano Materials Research Center, Kyungpook National University, Daegu 41566, Korea

**Keywords:** *Manilkara zapota* L., electrospray ionization (ESI), atmospheric pressure chemical ionization (APCI), antioxidant, α-glucosidase

## Abstract

High-resolution mass spectrometry equipped with electrospray ionization (ESI) and atmospheric pressure chemical ionization (APCI) sources was used to enhance the characterization of phytochemicals of ethanol extracts of *Manilkara zapota* L. leaves (ZLE). Sugar compounds, dicarboxylic acids, compounds of phenolic acids and flavonoids groups, and other phytochemicals were detected from the leaves. Antioxidant activity and inhibition potentiality of ZLE against α-glucosidase enzyme, and elastase enzyme activities were evaluated in in vitro analysis. ZLE significantly inhibited activities of α-glucosidase enzyme at a lower concentration (IC_50_ 2.51 ± 0.15 µg/mL). Glucose uptake in C2C12 cells was significantly enhanced by 42.13 ± 0.15% following the treatment with ZLE at 30 µg/mL. It also exhibited potential antioxidant activities and elastase enzyme inhibition activity (IC_50_ 27.51 ± 1.70 µg/mL). Atmospheric pressure chemical ionization mass spectrometry (APCI–MS) detected more *m*/*z* peaks than electrospray ionization mass spectrometry (ESI–MS), and both ionization techniques illustrated the biological activities of the detected compounds more thoroughly compared to single-mode analysis. Our findings suggest that APCI along with ESI is a potential ionization technique for metabolite profiling, and ZLE has the potential in managing diabetes by inhibiting α-glucosidase activity and enhancing glucose uptake.

## 1. Introduction

Nature is an abundant source of medicinal plants; various natural foodstuffs containing prospective bioactive components which support potential health care are traditionally used as herbal remedies for numerous life-threatening diseases [1]. Medicinal plants for their potential bioactive compounds are used for the ailment of numerous illness like inflammations, oxidative stresses, fatness and hypertensive conditions, diabetic mellitus, skin care, neurological complaints, and cancer, and to regulate expression of various genes [2]. Thus, the demand of discovering and developing new natural therapeutics is of growing interest to the researchers. Screening of plant extracts using different ionization techniques in mass spectrometry is one of the fastest and better ways for the discovery of new bioactive compounds [3].

*Manilkara zapota* L. is commonly referred to as sapodilla and also locally known as sofeda in Bangladesh, belonging to the Sapotaceae family. It is a long-living, evergreen plant commonly found in India, the south part of Mexico, and Central America [4]. Sapodilla is edible and has a sweet taste with rich flavor. There have been some studies on the fruits, seeds, barks, and leaves of *M. zapota* for phytochemical characterization and assessment of biological activities such as antioxidant, antimicrobial, antihyperglycemic, hypocholesterolemic, antityrosinase, and antidiabetic properties [4,5,6,7,8,9]. However, there have been few studies on thorough phytoconstituent identification based on advanced mass spectrometry and assessment of α-glucosidase enzyme inhibition activity of *M. zapota* leaves. Hence, characterization of bioactive secondary metabolites of *M. zapota* L. leaves and evaluation of their potential biological activities are of greater interest for further investigating the prospective use of sapodilla because of climacteric change, and as this becomes possible, the popularity of this fruit in many countries is rapidly increasing.

Mass spectrometry (MS)-based analytical methods are highly sensitive for the identification, quantification, and structural interpretation of metabolites. Sample ionization is one of the most important steps for mass spectrometry-based metabolomics. Electrospray ionization (ESI), atmospheric pressure chemical ionization (APCI), and electron impact ionization (EI) are typically used in the field of metabolomics [10]. ESI ionizes competently the strong and moderate polar metabolites, whereas APCI is typically considered for nonpolar and less-polar metabolites [11]. Mass spectrometers using different ionizations provide a mass-to-charge ratio (*m*/*z* peaks) of ions by generating mass spectrum, and tandem mass spectrometry (MS/MS) split precursor ions into smaller fragmented ions for the structural elucidation of unknown compounds [12,13].

In this study, direct injection negative-mode ESI–MS and APCI–MS were undertaken to explore a broad range of metabolites from *M. zapota* L. leaves. Furthermore, various in vitro tests were performed to assess its α-glucosidase inhibition activity, uptake of glucose in C2C12 myotubes, and its antioxidant and elastase inhibition activities. In this study, the experimental findings were also compared with those of other previous studies. Here, a systematic approach for metabolite profiling and bioactivity studies of ZLE was employed.

## 2. Results

### 2.1. Identification of Secondary Metabolites

Obtained mass spectra of the extracts from negative-mode ESI–MS and APCI–MS are displayed in Figure 1. Using negative-mode ESI–MS and APCI–MS, 174 and 225 peaks were detected, respectively; 65 peaks were commonly found with both ionization techniques (Appendix A). Peak lists detected from mass spectra in APCI–MS and ESI–MS are provided in Appendix A. Proposed compounds were first assumed by comparing their experimental masses in high-resolution mass spectrometry with theoretical calculated masses.

Identification of compounds was carried out using tandem mass spectrometry (MS/MS), which is an eminent technique to elucidate the structure of compounds and thereby very effective to identify unknown compounds comprising complex mixtures. MS/MS fragmentations of all targeted *m*/*z* peaks were carried out using the higher-energy collisional dissociation (HCD) activation mode at collision energy (CE) values of 10, 20, and 30 eV.

Gallic acid (*m*/*z* 169.0129), myricetin (*m*/*z* 317.0296), quinic acid (*m*/*z* 191.0548), chlorogenic acid (*m*/*z* 353.0866), and myricetin-3-*O*-rhamnoside (*m*/*z* 463.0868) peaks were observed abundant in negative-mode ESI–MS and APCI–MS spectra of ZLE (Figure 1). From ZLE analyzed in ESI–MS and APCI–MS, 39 compounds including 3 sugar compounds, 4 dicarboxylic acids, 8 phenolic acids, 9 flavonoids, and 15 other phytochemicals were identified (Table 1). Sugar compounds rhamnose, glucose, and sucrose were detected at *m*/*z* 163.0598, 179.0547, and 341.1077, respectively [14,15].

Succinic acid was detected at *m*/*z* 117.0179 [M–H]^−^ with fragmented peaks at *m*/*z* 99.0075 [M–H–H_2_O]^−^ and 73.0282 [M–H–CH_2_O_3_]^−^ (Appendix A) [16]. Malic acid was identified at *m*/*z* 133.0128 [M–H]^−^ with fragmented peaks at *m*/*z* 115.0024 [M–H–H_2_O]^−^, 89.0231 [M–H–CO_2_]^−^, and 71.0125 [M–H–CH_2_O_3_]^−^ (Appendix A) [17]. Adipic acid was proposed at *m*/*z* 145.0492 with fragmented peaks at *m*/*z* 127.0387 and 101.0594. 3-Oxoadipic acid was supposed at *m*/*z* 159.0285 with its yielded fragments at *m*/*z* 141.0180, 115.0387, and 97.0281 [15].

*m*/*z* 137.0230 was identified as salicylic acid as it produced fragments at *m*/*z* 93.0333. 3,4-Hihydroxybenzoic acid was suggested at *m*/*z* 153.0179 and it displayed fragment ion at *m*/*z* 109.0282 [18]. Vanillic acid, caffeic acid, ferulic acid, and syringic acid were detected at *m*/*z* 167.0336, 179.0336, 193.0591, and 197.0442, comparing the data found in the Human Metabolome Database (HMDB) and literatures [15,19,20]. *m*/*z* 169.0129 [M–H]^−^ was characterized as gallic acid and yielded a fragmented ion at *m/*z 125.0232 corresponding to [M–H–CO_2_]^−^ (Appendix A) [21].

Afzelechin was supposed at *m*/*z* 273.0765 [15]. Epicatechin was detected at *m*/*z* 289.0708 and it generated fragmented peaks at *m*/*z* 245.0812 [M–H–CO_2_]^−^, *m*/*z* 205.0497 [M–H–C_4_H_4_O_2_]^−^, *m*/*z* 179.0339 [M–H–C_6_H_6_O_2_]^−^, and *m*/*z* 137.0232 [M–H–C_8_H_8_O_3_]^−^ (Appendix A) [22]. Epigallocatechin had a peak at *m*/*z* 305.0656 and upon fragmentation exhibited peaks at *m*/*z* 287.0561, 261.0769, 219.0659, 179.0343, 165.0186, and 125.0235, respectively [23]. *m*/*z* 317.0296 [M–H]^−^ was proposed to be myricetin following fragmented peaks at *m*/*z* 193.0135, 165.0185, and 137.0234 through the loss of C_6_H_4_O_3_, C_7_H_4_O_4_, and C_8_H_4_O_5_, respectively (Appendix A). *m*/*z* 319.0447 exhibited fragmented ions at *m*/*z* 193.0137, 178.9980, 153.0185, and 125.0234, accordingly, and identified as ampelopsin [24]. *m*/*z* 331.0454 was characterized as laricitrin due to the appearance of fragmented ions at *m*/*z* 316.0219, 178.9976, and 151.0028, accordingly [25]. In addition, myricetin-3-O-rhamnoside was supposed at *m*/*z* 463.0868 with the fragmented ion at *m*/*z* 316.0217 [M–H–C_6_H_11_O_4_]^−^ (Figure 2A) [26]. *m*/*z* of 353.0866 [M–H]^−^ produced a fragmented peak at *m*/*z* 191.0551 at collision energy 10 eV with losing C_9_H_6_O_3_, and the compound was suggested as chlorogenic acid (Figure 2B) [27]. *m*/*z* 477.1022 [M–H]^−^ was suggested to be laricitrin-3-*O*-rhamnoside with fragmented ions of *m*/*z* 331.0453 [M–H–C_6_H_10_O_4_]^−^, *m*/*z* 316.0208 [M–H–C_6_H_10_O_4_–CH_3_]^−^, and *m*/*z* 287.0192 [M–H–C_6_H_10_O_4_–CH_3_–CHO]^−^ (Figure 2C) [25]. Prodelphinidin B was detected at *m*/*z* 609.1233 based on literature data with similar MS/MS fragmented ions at *m*/*z* 441.0818, 423.0711, 305.0659, 191.0550, and 125.0231, accordingly [28].

Compared to data from HMDB and the literature [15,16,18,20,29], *m*/*z* 121.0280, 123.0437, 128.0386, 135.0285, 151.0388, 161.0230, 173.0442, and 177.0179 were characterized as 2-Hydroxybenzaldehyde, guaiacol, pyroglutamic acid, threonic acid, vanillin, 3-Hydroxycumarin, shikimic acid, and esculetin. The peak at *m*/*z* 191.0548 [M–H]^−^ was supposed as quinic acid by detecting fragmented peaks at *m*/*z* 173.0445, 127.0388, 109.0282, and 93.0333 with the loss of H_2_O, CH_4_O_3_, CH_6_O_4_, and CH_6_O_5_, accordingly (Appendix A) [30]. *m*/*z* 259.0242 [M–H]^−^ was identified as norathyriol based on fragmentation peaks at *m*/*z* 231.0295, 215.0345, 187.0394, and 171.0444 with the loss of CO, CO_2_, C_2_O_3_, and C_2_O_4_ (Appendix A) [31]. *m*/*z* 285.0615, 321.0604, and 331.0655 were proposed to be hydroquinone glucuronide, leucodelphinidin, and 3-Glucogallic acid following the comparison with the HMDB database and literatures [15,32].

*m*/*z* 337.0917 showed fragmented ions at *m*/*z* 191.0555, 173.0449, and 163.0393 upon fragmentation at collision energy (CE) 10 eV and it was characterized as 3-*p*-Coumaroylquinic acid [20,33]. *m*/*z* 343.0658 was identified as 3-*O*-Galloylquinic acid as it produced fragmented ions at *m*/*z* 191.0555, 173.0449, 169.0136, and 125.0235, accordingly [15,34].

### 2.2. Antioxidant Effects of ZLE

In this study, antioxidant activity of ZLE was evaluated following DPPH^●^ and ABTS^●+^ scavenging assays, and the results of the assays are shown in Figure 3A,B. It was found that the percentage inhibition of DPPH^●^ was 25.31%, 36.07%, 56.06%, 80.21%, and 86.28% at 1, 3, 10, 30, and 100 µg/mL, accordingly. There was a progressive radical scavenging trend with a higher concentration of ZLE. IC_50_ of ZLE was determined at 7.93 ± 1.43 µg/mL, where ascorbic acid had IC_50_: 5.34 ± 0.27 µg/mL, which indicated that the leaf extracts possessed a powerful DPPH^●^ scavenging activity.

Experimental results of the ABTS^●+^ scavenging assay are displayed in Figure 3B. At 100 µg/mL, the inhibition of ABTS^●+^ by the leaf extracts was 56.15% and IC_50_ was 72.85 µg/mL, which suggested that ZLE had a capacity to donate both electrons and hydrogen, resulting in having antioxidant potential. With the purpose of determining the reducing potentiality of ZLE, ferric-reducing antioxidant power (FRAP) and cupric-reducing antioxidant capacity (CUPRAC) assays were conducted. ZLE (100 µg/mL) demonstrated strong reducing power with 53.30 ± 2.85 µM ascorbic acid equivalent FRAP and 40.09 ± 3.61 µM ascorbic acid equivalent CUPRAC value, respectively. These results suggested that ZLE also had a strong electron donating capacity, resulting in having powerful antioxidant capacity (Figure 3C,D).

### 2.3. α-Glucosidase Inhibitory Activity of ZLE

α-Glucosidase enzymes aid in hydrolysis of carbohydrates including starches during digestion into glucose in the intestine which, in turn, contribute to the increase of blood glucose levels. Inhibiting the α-glucosidase enzyme activity with type 2 diabetes patients may suppress postprandial hyperglycemia [35]. ZLE’s ability to inhibit α-glucosidase is presented in Figure 4A and the results showed that ZLE inhibited the activity of α-glucosidase in concentration dependently. ZLE significantly inhibited α-glucosidase activity even at 1 µg/mL and the calculated IC_50_ was found very low (2.51 ± 0.15 µg/mL), whereas acarbose had a higher value (IC_50_ 216.26 ± 1.63 µg/mL).

### 2.4. Effects of ZLE on Cell Viability and Glucose Uptake in C2C12 Myotubes

The effects of ZLE on cell viability in C2C12 cells are presented in Figure 4B. ZLE exhibited moderate to high cytotoxicity at 100 and 200 μg/mL, respectively, whereas at up to 50 μg/mL, there were no cytotoxic effects observed. Glucose uptake potential of ZLE in C2C12 myotubes was measured following the treatment with ZLE at the designated time-points (Figure 4C). The effect of ZLE (at 30 μg/mL) on basal- or insulin-induced glucose uptake initiated at 0.5 h, which was utmost at 2 h then gradually reduced. Hence, we considered 2 h time period for ZLE administration in the sequential assays. We observed that basal- or insulin-administered glucose uptake augmented in a concentration proportional fashion (Figure 4D). Basal- and insulin-administered glucose uptake improved remarkably by 42.13 ± 0.27% and 57.74 ± 0.17%, respectively, at the highest concentration of ZLE (30 μg/mL).

Rosiglitazone, a commonly recognized thiazolidinedione group antidiabetic agent, was applied here in the form of positive control. It also substantially augmented basal- and insulin-applied glucose uptake levels by 28.49 ± 0.49% and 50.38 ± 0.32%, respectively.

### 2.5. Elastase Inhibition Activity of ZLE

Experimental results showed that elastase inhibition activity was increased with the increased concentration of ZLE (Appendix A). IC_50_ for elastase inhibition activity was measured at a concentration of 27.51 ± 1.70 µg/mL of ZLE, whereas less inhibition activity was observed for EGCG at a higher concentration.

## 3. Discussion

Previous studies found that ethanol extracts of plant materials exhibit good biological activities and contain more bioactive components [36,37,38]. In this study, we performed ethanol extraction of *M. zapota* leaves (ZLE) and used ZLE for the subsequent experiments.

Mass spectrometry analysis employing direct sample solution injection is a rapid and cost-effective technique compared to other separation procedures [39]. Determination of precise mass and its fragmentation peak using tandem mass spectrometry (MS/MS) aids in the elemental composition information for unknown compound identification. However, for accurate and isomeric identification, we needed to follow a sequential analysis using liquid chromatography mass spectrometry (LC MS), infrared (IR) spectroscopy, and nuclear magnetic resonance (NMR) spectroscopy [40,41].

Today, the ESI technique is widely used for the analysis of singly-charged small molecules to multiply charged large molecules in complex substances [42,43]. APCI is also applicable for the analysis of less polar compounds with a molecular weight of <1500 Da [44]. In this study, direct injection APCI was used along with the ESI technique equipped with high-resolution MS for the phytochemical screening of ZLE. We used negative-mode electrospray ionization (ESI) in mass spectrometry for the metabolite profiling of *M. zapota* leaves because it yielded enhanced sensitivity and more observable peaks in the mass spectra [45]. Negative-mode ESI-MS presented extensive insights of polyketide classes (flavones, flavanols, flavanones, isoflavones, flavonols, phloroglucinols, anthraquinones, bisanthraquinones, and stilbenes) compounds particularly through the loss of H_2_O, CO, CH_2_CO, and CO in tandem mass spectrometry (MS/MS) [46].

In this study, more *m*/*z* peaks were observed in the APCI technique (225 peaks) than ESI technique (174 peaks), and 65 *m*/*z* peaks were found common. Using both ESI and APCI, 334 *m*/*z* peaks of ZLE were detected, which specified the advantage of using both techniques for enhanced metabolite profiling. As a result, a higher number of compounds could be identified in this study compared to previous ones. One study reported the identification of six compounds, including apigenin-7-*O*-α-ʟ-rhamnoside, oleanolic acid, caffeic acid, lupeol acetate, myricetin-3-*O*-α-ʟ-rhamnoside, some hydrocarbons, and fatty acids [5], and another study reported the presence of myricetin-3-*O*-α-ʟ-rhamnoside and myricetin [47] in the *M. zapota* leaves. However, in this study, 39 compounds including sugar, dicarboxylic acids, phenolic acids, flavonoids, and other phytochemicals were identified in ZLE (Table 1).

DPPH^●^ and ABTS^●+^ scavenging assays are widely practiced procedures for the determination of antioxidant prospective of natural supplements (e.g., foods, plants). DPPH^●^ and ABTS^●+^ scavenging methods measure the transfer of hydrogen or electrons from potential antioxidants to free radicals. In this study, ZLE exhibited more significant DPPH^●^ scavenging activity than the ABTS^●+^ scavenging activity (Figure 3A,B). FRAP and CUPRAC assays were performed to measure the reducing potentiality of antioxidants in which the reduction is performed with the assistance of compounds which break free-radical chain through donating hydrogen atom [48]. Experimental findings suggest that ZLE exhibited a noticeable antioxidant potential (ascorbic acid equivalent CUPRAC and FRAP values), increased with increasing concentration of the samples (Figure 3C,D). Taken together, the results demonstrated that ZLE has a strong potential antioxidant activity. The DPPH^●^ and ABTS^●+^ scavenging abilities of *Manilkara zapota* fruit with IC_50_ 37.63 ± 1.18 μg/mL and 73.14 ± 2.84 μg/mL, respectively, have been reported [49]. However, in our present study, we found the DPPH^●^ and ABTS^●+^ scavenging activities of ZLE with IC_50_ 7.93 ± 1.43 and 72.85 μg/mL, accordingly, which indicated that *M. zapota* leaves might be a promising bioactive agent.

Phenolic compounds exhibit antioxidant activities through donating hydrogen to the reactive oxygen or nitrogen species from the hydroxyl groups, which broke the radical generating cycle [50]. Identified phenolic compounds of *M. zapota* leaves such as 3,4-dihydroxybenzoic acid (DPPH^●^: IC_50_ 1.88 µg/mL, ABTS^●+^: IC_50_ 0.89 µg/mL) [51], gallic acid (DPPH^●^: IC_50_ 3.53 ± 0.24 µg/mL, ABTS^●+^: IC_50_ 8.85 ± 0.74 µg/mL), caffeic acid (DPPH^●^: IC_50_ 6.34 ± 0.37 µg/mL, ABTS^●+^: IC_50_ 18.04 ± 0.68 µg/mL), syringic acid (DPPH^●^: IC_50_ 5.44 ± 0.53 µg/mL, ABTS^●+^: IC_50_ 13.97 ± 0.59 µg/mL), ferulic acid (DPPH^●^: IC_50_ 11.75 ± 0.45 µg/mL, ABTS^●+^: IC_50_ 9.47 ± 0.69 µg/mL) [52], ampelopsin (DPPH^●^: IC_50_ 4.94 µM), and myricetin-3-*O*-rhamnoside (DPPH^●^: IC_50_ 5.14 µM) [53] have been reported to possess antioxidant activity.

More research is focusing on the functional food field especially on foods, and plants to prevent type-2 diabetes because of their α-glucosidase inhibition activity [54]. Various phytochemicals from herbs, barks, fruits, leaves, and different parts of medicinal plants have been recorded to exhibit inhibition against α-glucosidase activity and used as a remedy for type-2 diabetes [55]. Dichloromethane extracts of *Croton bonplandianum* [56]; ethanol extracts of *Enhalus acoroides*, *Thalassia hemprichii*, and *Cymodocea rotundata* [57]; ethanol extracts of *Castanea mollissima* [58]; acetone extracts of *Undaria pinnatifida* [59]; ethanol extracts of *Vitis aestivalis* [60]; and water extracts and ethanol extracts from *Fucus vesiculosus* [61] exhibited α-glucosidase inhibition abilities with IC_50_ values of 14.93, 168.15, 425.86, 429.28, 2.3, 80, 384, 0.32, and 0.49 µg/mL, respectively, and have traditionally been used for treatment of type 2 diabetes (T2D). Distilled water extraction of *Achras sapota* (*M. zapota*) fruit has been reported to inhibit α-glucosidase enzyme with IC_50_ 56 µg/mL [62], whereas our results revealed that ZLE had significant potential for inhibiting α-glucosidase activity with IC_50_ 2.51 ± 0.15 µg/mL, which suggested that ZLE might be as a prospective source in developing anti-diabetic agents.

It has been reported that caffeic acid and chlorogenic acid (compounds found in ZLE) have remarkable α-glucosidase inhibition ability with IC_50_ values of 4.98 and 9.24 µg/mL, respectively [63]. Other phenolic compounds like ferulic acid (IC_50_ 10.80 ± 0.90 mg/mL) [64] and epicatechin (IC_50_ 5.86 µg/mL) [65] also possess α-glucosidase inhibition activity. In this study, we identified norathyriol and myricetin from ZLE using negative-mode APCI–MS which possesses a potential α-glucosidase inhibition activity with IC_50_ values of 0.81 and 2.09 µg/mL, accordingly [66,67]. Therefore, it was assumed that ZLE showed a significant α-glucosidase inhibition potential due to the combined effect of these compounds.

As we found that ZLE possesses significant α-glucosidase inhibition activity, we carried out the uptake of glucose in C2C12 myotubes for determining the prospects of ZLE as an anti-diabetic agent. Skeletal muscle is an important site for the insulin-mediated glucose uptake, transport, and repository. Furthermore, an increase in glucose uptake in this cell is considered to play a key function in the control of glucose homeostasis and regulation of T2D [68]. ZLE significantly improved the uptake of glucose in C2C12 cells in concentration dependently. We observed synergistic effect of ZLE on glucose uptake in C2C12 cells on exposure to insulin, which suggested the role of ZLE to imitate the insulin-like actions (Figure 4D).

Collagen and elastin being the prominent proteins in the body modulate important biological and mechanical functions. Collagen is the principal part of the extracellular matrix (ECM), and it provides structural support and strength. On the other hand, elastin is the most prevalent constituent of ECM, which contributes to elasticity in living tissues [69,70,71]. Elastin is degraded by the activation of elastase upon UV irradiation [72]. We found in this study that ZLE strongly inhibited elastase enzyme activity (Appendix A). In our previous study, it was found that different combinations of epicatechin, syringic acid, and vanillic acid had a substantial interactive effect on inhibitory ability of elastase [73]. Sirinya et al. found the elastase inhibition activity of *Manilkara zapota* fruit with IC_50_ 35.73 ± 0.61 μg/mL [49], whereas in our study ZLE exhibited elastase inhibitory action with IC_50_ 27.51 ± 1.70 µg/mL. Therefore, *M. zapota* leaves possessed more significant elastase inhibition activity than the fruits.

## 4. Materials and Methods

### 4.1. Chemicals

Neocuproine (≥98%), *p*-nitrophenyl-α-D-glucopyranoside (pNPG) (≥99%), 2,2-diphenyl-1-picrylhydrazyl (DPPH) (≥ 98%), 2,4,6-tris(2-pyridyl)-*s*-triazine (≥99%), 2,2′-azino-bis(3-ethylbenzthiazoline-6-sulphonic acid (ABTS) (≥98%), α-glucosidase (23 units/mg protein), N-succinyl-(Ala)-3-*p*-nitroanilide (≥98%), dimethylsulfoxide (DMSO), ascorbic acid (≥99%), elastase (7.6 units/ mg protein), and HPLC-grade ethanol were obtained from Sigma–Aldrich (St. Louis, MO, USA). We used all of these chemicals without further purification to conduct this research work.

### 4.2. Plant Materials and Extraction

Leaves of *M. zapota* L. were collected from the Khulna area of Bangladesh in June 2018. Taxonomic identification (accession number DACB-23801) was carried out at the Bangladesh National Herbarium. Shade-dried fresh leaves were ground into fine homogenized powder using a mortar and pestle. Of the fine power, 100 g was processed in three cycles with a reflux system of the extraction process of 6 h using ethanol. A Whatman filter paper (Schleicher and Schuell, Keene, NH, USA) was used to filter the extracts. Evaporation of the solvent was done using a rotary evaporator (SB-1000, Tokyo Rikakikai Co. Ltd., Tokyo, Japan). At last, we obtained powdered extracts of the sample after using a freeze-dryer (MCFD 8518, Ilshin Biobase Co. Ltd., Goyang, Korea).

### 4.3. Sample Preparation

Stock solution of ZLE was prepared by dissolving the extract in 100% ethanol (10 mg/mL), and subsequently, the stock solution was diluted using ethanol and water for MS analysis. Homogenization of the prepared samples solution was made following 1 min vortex and 5 min sonication in a sonication bath (Powersonic 410, Hwashin Technology Co., Gyeonggi, South Korea). For in vitro analysis, the extracts at a concentration of 30 mg/mL were dissolved in DMSO and deionized water (50:50, *v*/*v*), and subsequent diluted solutions were made using deionized water.

### 4.4. Mass Spectrometry Analysis

Negative-mode ESI–MS and APCI–MS experiments were executed with Q-Exactive™ Orbitrap Mass Spectrometer (Thermo Fisher Scientific Inc., San Jose, CA, USA). A 500-µL syringe (Hamilton Company Inc., Reno, NV, USA) and a syringe pump (model 11, Harvard, Holliston, MA, USA) were used to infuse the sample solution into the ESI source at 20 µL/min. Typical negative-mode ESI–MS conditions were at mass resolution 140,000 (full width at half maximum, FWHM), sheath gas flow rate 6, auxiliary gas flow rate 2, sweep gas flow rate 0 (arbitrary units), spray voltage 4.00 kV, capillary temperature 400 °C, S-lens RF level 50, and automatic gain control 5 E^6^. Highly pure (99.99 %) nitrogen obtained from the evaporation of liquid nitrogen was applied as the sheath, auxiliary, and sweep gas. Negative-mode external calibration was performed using Pierce Velos solution (Thermo Fisher Scientific, Rockford, IL, USA) into the ESI interface.

Operating conditions for negative-mode APCI–MS were as follows: sample infusion rate 200 µL/min through a 5-mL syringe (Hamilton Company Inc., Reno, NV, USA), mass resolution (FWHM) 140,000, sheath gas flow rate 15, auxiliary gas flow rate 2, sweep gas flow rate 0 (arbitrary units), discharge current 2 µA, capillary temperature 300 °C, vaporizer temperature 400 °C, S-lens RF level 70, and automatic gain control 5 E^6^.

Mass spectra were acquisitioned from *m*/*z* 50 to *m*/*z* 750 for both ionizations and 10–30 eV collision energy was applied in the tandem mass spectrometry (ESI-MS/MS and APCI-MS/MS).

Experimental conditions for ESI and APCI mass spectrometry were optimized considering the most abundant peak’s intensity to obtain a good quality spectrum. Operating parameters for ESI–MS and APCI–MS were optimized and briefed in the Appendix A.

### 4.5. Data Processing

Obtained mass spectrometry data were handled using Xcalibur 3.1 with Foundation 3.1 (Thermo Fisher Scientific, USA). A signal-to-noise ratio (S/N) value of 100 was considered to detect *m*/*z* peaks, and exported peak lists were further processed using Microsoft Excel. Elemental formulas for the detected *m*/*z* peaks were assigned within typical conditions of C*_c_*H*_h_*N*_n_*O*_o_*, counting *c* unlimited, *h* unlimited, and 0 ≤ *n* ≤ 10, and 0 ≤ *o* ≤ 50 using MIDAS Formula Calculator. ChemDraw Professional 15.0 (PerkinElmer, Waltham, MA, USA) was used to draw the structures of the identified compounds. A Venn diagram for the overlapped peaks was generated by FunRich 3.1.3 [74].

Compound identification was executed by comparing theoretically calculated masses of [M–H]^−^ and [M+Cl]^−^ adducts with the observed *m*/*z* peak values. For the structural elucidation of the detected phytochemicals, ESI–MS/MS- and APCI–MS/MS-produced fragments of the target peaks were compared with the Human Metabolome Database (online database) [15] and literatures.

### 4.6. Antioxidant Activity Assays

The DPPH^●^ scavenging capability of ZLE was determined following a previous methodology [75]. Into 2-µL sample solutions of different concentrations (1, 3, 10, 30, and 100 µg/mL), 198 µL of 0.2 mM solution of DPPH in 50% ethanol was added. The mixture was allowed to stand for 10 min at 25 °C, and the absorbance was recorded by a microplate reader (Multiskan GO 51119200; Thermo Fisher Scientific Oy, Ratastie, 01620 Vantaa, Finland) at 517 nm. Ascorbic acid was used as a reference antioxidant compound.

The ABTS^●+^ scavenging activity was determined adapting the method described by Re et al. [76] with slight modifications, and 2 µL of different concentrations of ZLE (1, 3, 10, 30, and 100 µg/mL) was mixed with 198 µL ABTS solution, and the absorbance was recorded at 734 nm. Ascorbic acid was used as a reference antioxidant compound, and both DPPH^●^ and ABTS^●+^ scavenging activities were calculated using Equation (1).
Radical-scavenging activity (% inhibition) = [(A_control_ − A_sample_) / A_control_] × 100(1)
where A_control_ is the control’s absorbance and A_sample_ is the sample’s absorbance. Data were obtained for three samples.

The ferric-reducing antioxidant power (FRAP) assay and cupric-reducing antioxidant capacity (CUPRAC) assay were performed following the reported methods [77,78]. Then, 2 µL of the sample solution of ZLE at different concentrations (1, 3, 10, 30, and 100 µg/mL) was mixed with 198 µL of the FRAP reagent, and the absorbance was measured at 595 nm. Ascorbic acid was also used as a reference compound. In the case of the CUPRAC assay, the samples at different concentrations (1, 3, 10, 30, and 100 µg/mL) were added with the mixture of 10 mM CuCl_2_, 7.5 mM neocuproine, and 1 M ammonium acetate buffer (pH 7.0). The mixture was incubated for 1 h at 25 °C temperature, and the absorbance was determined at 450 nm. Ascorbic acid was used as a reference compound, and the reducing capability was marked as ascorbic acid equivalent (in micromolar).

### 4.7. Inhibitory Assay of α-Glucosidase

The capability of ZLE to restrain α-glucosidase activity was determined in accordance with the reported method [79]. With 0.2 U/mL of α-glucosidase in 0.1 M sodium phosphate buffer (pH 7.0), the sample solutions of 2 µL of different concentrations (1, 3, 10, 30, and 100 µg/mL) were mixed, and incubated for 10 min at 37 °C. Then, pNPG (substrate) was mixed into the prepared solution to terminate enzyme–substrate activity at 37 °C for 1 h. The enzyme–substrate reaction was carried out at 37 °C for 30 min. Absorbance of the mixture was figured out utilizing a 96-well plate microplate reader (Multiskan GO 51119200; Thermo Fisher Scientific Oy, Ratastie, 01620 Vantaa, Finland) at 405 nm. Acarbose as a positive control was employed and inhibitory percentage was calculated following the Equation (2).
Inhibition of enzyme activity (%) = [(A_control_ − A_sample_)/A_control_] × 100(2)
where A_control_ is the control’s absorbance and A_sample_ is the sample’s absorbance. Three sets of the sample’s data were obtained.

### 4.8. Cell Culture and Cell Viability Assay

C2C12 myotubes (American Type Culture Collection, Manassas, VA, USA) were grown in Dulbecco’s modified eagle’s medium (DMEM) with 10% fetal bovine serum (FBS) and streptomycin-penicillin (100 µg/mL and 100 U/mL, accordingly; Hyclone, Mordialloc, VIC, Australia) at 37 °C under a 5% CO_2_ atmosphere. The viability of C2C12 cells were detected following the tetrazolium dye colorimetric test (MTT) assay. At first C2C12 cells were grown at a density of 1 × 10^5^ cells/well in 96-well plates for 24 h. They were treated with different concentrations of ZLE after reaching 90% confluence. After 24 h of incubation, an MTT reagent was added to each well and the plate was incubated for 1 h at 37 °C. The media was removed and then the wells were washed two times with phosphate-buffered saline (PBS) (pH 7.4). The intracellular formazan was solubilized in 100% DMSO. The absorbance was recorded by microplate reader (Perkin Elmer, Wallac Victor 3, MA, USA) at 570 nm and the viability percentage was determined.

### 4.9. Muscle Differentiation and Glucose Uptake Assay

C2C12 cells were grown at a density of 1 × 10^5^ cells/well in 96-well plates with DMEM supplemented with 10% fetal bovine serum (FBS) and 1% streptomycin–penicillin at 37 °C under 5% CO_2_ atmosphere. The cells were differentiated in DMEM containing 2% horse serum for 5 days after reaching confluence. The cells were allowed to starve in low glucose serum-free DMEM for 24 h. Thereafter, 2-deoxy-2-[(7-nitro-2,1,3-benzoxadiazol-4-yl) amino]-D-glucose (2-NBDG) assay was carried out to determine glucose uptake [80]. Briefly, cells were treated with different concentrations of ZLE and insulin (100 nM) for 30 min, followed by 20 µM of 2-NBDG for 6 h. The cells were rinsed three times using cold PBS after incubation and uptake of 2-NBDG uptake recorded with a microplate reader (Perkin Elmer, Wallac Victor 3, MA, USA) at an excitation wavelength of 490 nm and emission wavelengths of 535 nm, accordingly.

### 4.10. Elastase Inhibition Assay

ZLE’s potentiality to hinder elastase activity was employed based on a previous study [73]. The reaction was performed in a mixture of 0.04 U elastase and 0.1 M Tris–HCl buffer (pH 8.0) with 0.78 mM *N*-succinyl-(Ala)-3-*p*-nitroanilide (substrate). With 100 µL of the substrate solution, different concentrations of the sample solution (1, 3, 10, 30, and 100 µg/mL) were mixed and then added to 100-µL enzyme solution. Then, mixer absorbance was figured out using a microplate reader (Multiskan GO 51119200; Thermo Fisher Scientific Oy, Ratastie, 01620 Vantaa, Finland) at 405 nm. Epigallocatechin gallate (EGCG) as a positive control was used, and the elastase inhibition percentage was calculated following above mentioned Equation (2).

### 4.11. Statistical Analysis

All results were as mean ± standard deviation. Linear regression analyses were carried out to calculate half maximal inhibitory concentration values (IC_50_). Statistical analyses were accomplished through one-way analysis of variance (ANOVA), following Dennett’s test by SigmaPlot 12.5 (Systat Software Inc., San Jose, CA, USA). Differences were counted significant if *p* < 0.05.

## 5. Conclusions

In this study, we found that combining ESI and APCI techniques with mass spectrometry maximized the coverage of *m*/*z* peak detection rather than using a single ionization technique. The combination of ESI and APCI ionization techniques in mass spectrometry provided us with more detailed characteristic features of metabolites and biological activities of natural compounds. We rapidly determined the secondary metabolites of *M. zapota* leaves using direct injection mode mass spectrometry. We identified 39 different phytochemicals of *M. zapota* leaves using negative-mode ESI–MS and APCI–MS. It was found that the leaf extracts of *M. zapota* had a significant α-glucosidase enzyme inhibition activity and also dose-dependently augmented glucose uptake alone or in combination with insulin in C2C12 myotubes. *M. zapota* leaves extracts also showed strong antioxidant and elastase inhibition potential. Compounds identified using negative-mode APCI–MS together with negative-mode ESI–MS strongly demonstrated the α-glucosidase activity of ZLE. It can be concluded that combining ESI and APCI with high-resolution mass spectrometry is an enhanced approach for metabolite profiling and biological activity studies. The findings of this work suggest that *M. zapota* leaves can be a potential resource for antidiabetic agents, antioxidants, and elastase enzyme inhibitors. Furthermore, we need to carry out liquid chromatography mass spectrometry (LC-MS) to determine the concentration of bioactive compounds. In vivo biological assays are also needed to be performed for the better characterization of antidiabetic ability of *M. zapota* leaves. Finally, we are interested to continue research on selective bioactive compounds for enhanced antidiabetic activity.

## Figures and Tables

**Figure 1 ijms-22-00132-f001:**
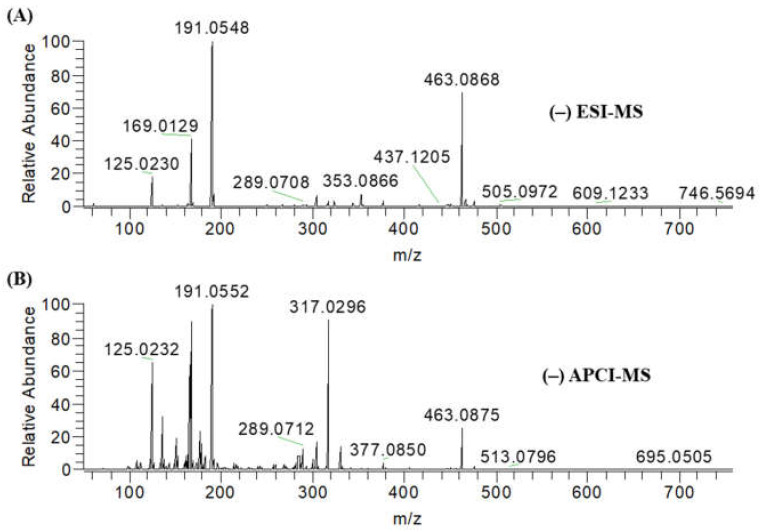
(**A**) Negative-mode electrospray ionization mass spectrometry (ESI–MS), (**B**) negative-mode atmospheric pressure chemical ionization mass spectrometry (APCI–MS) mass spectra of ethanol extracts of *M. zapota* leaves (ZLE). Prepared sample was directly infused to ESI and APCI sources.

**Figure 2 ijms-22-00132-f002:**
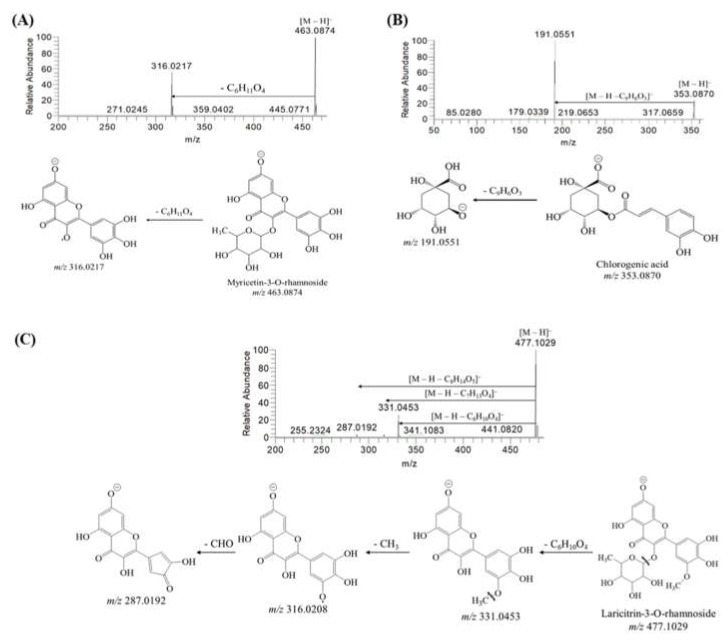
Negative-mode electrospray ionization tandem mass spectrometry (ESI–MS/MS) spectra and schematic fragmentation patterns of (**A**) myricetin-3-*O*-rhamnoside, (**B**) chlorogenic acid, and (**C**) laricitrin-3-*O*-rhamnoside; 10–30 eV collision energy (CE) was applied to produce fragmented peaks of the precursor ions.

**Figure 3 ijms-22-00132-f003:**
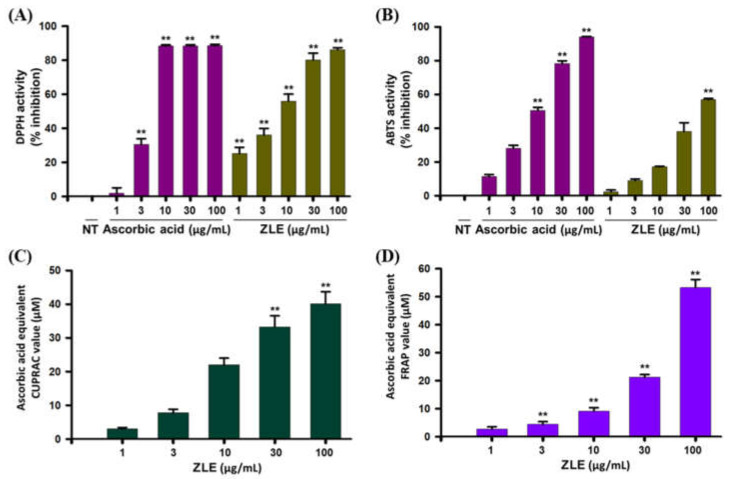
Antioxidant potential of ethanol extracts of *M. zapota* L. leaves (ZLE). (**A**) 2,2-Diphenyl-1-picrylhydrazyl radical (DPPH•) scavenging, (**B**) 2,2′-Azino-bis(3-ethylbenzthiazoline-6-sulphonic acid radical cation (ABTS•+) scavenging, (**C**) Cupric-reducing antioxidant capacity (CUPRAC), and (**D**) Ferric-reducing antioxidant power (FRAP) assays. Ascorbic acid as a positive control was used in the assays. Results were as mean ± standard deviation of three experiments (** *p* < 0.05 versus control using one-way ANOVA).

**Figure 4 ijms-22-00132-f004:**
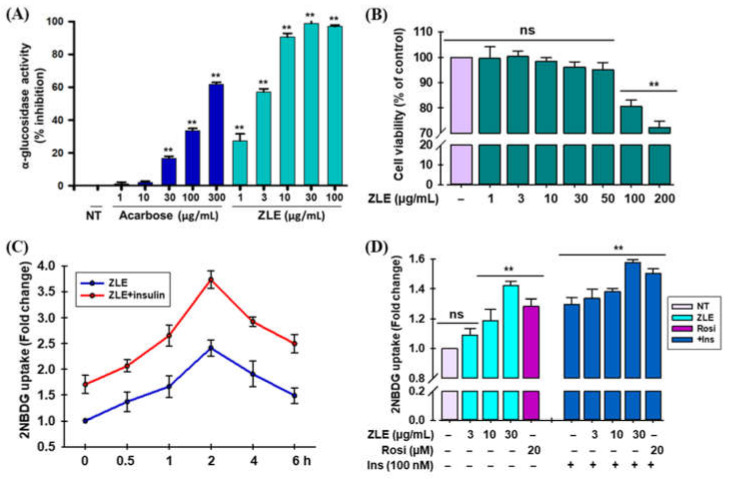
(**A**) α-Glucosidase inhibition activity of ethanol extracts of *M. zapota* leaves (ZLE). Acarbose as positive control was used in the assay. (**B**) Cell viability in C2C12 myotubes. C2C12 cells were seeded in a 96-well plate (density 1 × 10^5^ cells/well) and carried out 3-(4,5-dimethylthiazol-2-yl)-2,5-diphenyltetrazolium bromide (MTT) assay as referred to in Materials and Methods. (**C**) Time-dependent uptake of glucose by C2C12 myotubes. Cells treated with ZLE (30 μg/mL) alone or ZLE (30 μg/mL) along with insulin (100 nM) were incubated for the time periods indicated. (**D**) Effects of ZLE on the uptake of glucose in C2C12 myotubes. Cells were incubated with ZLE alone and/or with insulin for 2 h. Rosiglitazone (20 μM) was employed as a positive control. (‘ns’ indicated non-significant; ‘–‘ indicated sample was not added and ‘+’ indicated sample was added). Results were as mean ± standard deviation of three analyses (** *p* < 0.05 versus control using one-way ANOVA).

**Table 1 ijms-22-00132-t001:** Secondary metabolites identified in leaves of *Manilkara zapota* using negative-mode electrospray ionization mass spectrometry (ESI–MS) and atmospheric pressure chemical ionization mass spectrometry (APCI–MS).

Group	Compound Name	Elemental Formula	Observed *m*/*z*	Calculated *m*/*z*	Adducts	MS/MS Fragments	CE * (eV)	Ionizations
**Sugar**	Rhamnose	C_6_H_12_O_5_	163.0598	163.0611	[M–H]^−^	145, 119	20	ESI, APCI
	Glucose	C_6_H_12_O_6_	179.0547	179.0561	[M–H]^−^	161, 143, 113, 101	10	ESI, APCI
			215.0316	215.0324	[M+Cl]^−^	179, 161, 143, 113, 101	10	ESI
	Sucrose	C_12_H_22_O_11_	341.1077	341.1083	[M–H]^−^	179, 161, 119, 101	10	ESI, APCI
**Dicarboxylic acid**	Succinic acid	C_4_H_6_O_4_	117.0179	117.0193	[M–H]^−^	99, 73	10	APCI
	Malic Acid	C_4_H_6_O_5_	133.0128	133.0133		115, 89, 71	10	ESI, APCI
	Adipic acid	C_6_H_10_O_4_	145.0492	145.0506		127, 101	10	APCI
	3-Oxoadipic acid	C_6_H_8_O_5_	159.0285	159.0298		141, 115, 97	10	APCI
**Phenolic acids**	Salicylic acid	C_7_H_6_O_3_	137.0230	137.0244	[M–H]^−^	93	20	ESI, APCI
	3,4-Dihydroxybenzoic acid	C_7_H_6_O_4_	153.0179	153.0193		109	10	ESI, APCI
	Vanillic acid	C_8_H_8_O_4_	167.0336	167.0349		152, 123, 108	10	ESI, APCI
	Gallic acid	C_7_H_6_O_5_	169.0129	169.0142		125	10	ESI, APCI
	Caffeic acid	C_9_H_8_O_4_	179.0336	179.0349		161, 135	10	ESI, APCI
	Ferulic acid	C_10_H_10_O_4_	193.0591	193.0506		178, 149, 134	10	ESI, APCI
	Syringic acid	C_9_H_10_O_5_	197.0442	197.0455		182, 153, 125	10	ESI, APCI
	Chlorogenic acid	C_16_H_18_O_9_	353.0866	353.0878		191	10	ESI, APCI
**Flavonoids**	Afzelechin	C_15_H_14_O_5_	273.0765	273.0768	[M–H]^−^	167	10	APCI
	Epicatechin	C_15_H_14_O_6_	289.0708	289.0719		245, 205, 179, 137	20	ESI, APCI
	Epigallocatechin	C_15_H_14_O_7_	305.0656	305.0666		287, 261, 219, 179	10	ESI, APCI
	Myricetin	C_15_H_10_O_8_	317.0296	317.0302		193, 165, 137	20	APCI
	Ampelopsin	C_15_H_12_O_8_	319.0447	319.0459		193, 178, 153, 125	10	ESI
	Laricitrin	C_16_H_12_O_8_	331.0454	331.0459		316, 178, 151	20	ESI, APCI
	Myricetin-3-*O*-rhamnoside	C_21_H_20_O_12_	463.0868	463.0882		316	30	ESI, APCI
	Laricitrin-3-*O*-rhamnoside	C_22_H_22_O_12_	477.1022	477.1038		331, 316, 287	20	ESI, APCI
	Prodelphinidin B	C_30_H_26_O_14_	609.1233	609.1249		441, 423, 305, 125	10	ESI
**Others**	2-Hydroxybenzaldehyde	C_7_H_6_O_2_	121.0280	121.0295	[M–H]^−^		30	ESI, APCI
	Guaiacol	C_7_H_8_O_2_	123.0437	123.0451		108, 93	10	APCI
	Pyroglutamic acid	C_5_H_7_NO_3_	128.0386	128.0353		82	20	ESI, APCI
	Threonic acid	C_4_H_8_O_5_	135.0285	135.0298		117, 91, 75	10	ESI, APCI
	Vanillin	C_8_H_8_O_3_	151.0388	151.0407		136	10	ESI, APCI
	3-Hydroxycoumarin	C_9_H_6_O_3_	161.0230	161.0244		133, 117	10	ESI, APCI
	Shikimic acid	C_7_H_10_O_5_	173.0442	173.0455		155, 129	10	ESI, APCI
	Esculetin	C_9_H_6_O_4_	177.0179	177.0193		159, 149, 133, 121	10	ESI, APCI
	Quinic acid	C_7_H_12_O_6_	191.0548	191.0561		173, 127, 93, 85	20	ESI, APCI
	Norathyriol	C_13_H_8_O_6_	259.0242	259.0247		231, 215, 187, 171	30	APCI
	Hydroquinone glucuronide	C_12_H_14_O_8_	285.0615	285.0615		152, 109, 108	10	ESI
	Leucodelphinidin	C_15_H_14_O_8_	321.0604	321.0616		303, 195, 125	10	ESI
	3-Glucogallic acid	C_13_H_16_O_10_	331.0655	331.0670		287, 241, 169, 125	10	ESI
	3-*p*-Coumaroylquinic acid	C_16_H_18_O_8_	337.0917	337.0928		191, 173,163	10	ESI
	3-*O*-Galloylquinic acid	C_14_H_16_O_10_	343.0658	343.0670		191,173,169, 125	10	ESI
	Unknown	C_6_H_6_O_3_	125.0230	125.0244	[M–H]^−^	96	10	ESI, APCI
	Unknown	C_24_H_20_O_10_	467.0967	467.0983		357, 303, 217	10	ESI, APCI

Compounds were identified through comparison of their observed mass values with the calculated values and also comparing with the online database and literature. * CE: Collision energy. We used 10–30 eV collision energy for tandem mass spectrometry (MS/MS). For most of the cases, fragmentations of the precursor peaks occurred well at collision energy of 10 eV. The fragmented peak’s abundance started to increase with the increasing value of CE. We considered the optimal CE value for MS/MS fragmentations of the target peaks.

## Data Availability

The data presented in this study are available on request from the corresponding author.

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
