# Peer review of "Metabolite Profiling of Manilkara zapota L. Leaves by High-Resolution Mass Spectrometry Coupled with ESI and APCI and In Vitro Antioxidant Activity, α-Glucosidase, and Elastase Inhibition Assays"

_ijms, 2020, doi:10.3390/ijms22010132_

Round 1

Reviewer 1 Report

= doubts on the identification =

'Confirmed' or 'identified' or similar words in this context are not acceptable.

Until you do not have substantial evidence of all your peaks origin - it is only an approximation. Precise, but an approximation.

All approximations must be clearly stated to the readers (some of the readers are not deeply-involved in analytical methods and will uncritically repeat your "results").

e.g. 1

Explain clearly how is it possible to absolutely assign an [M–H]– ion, e.g., 463.0868 m/z, to be derived from myricetin-3-O-rhamnoside and exclude other deoxyhexoside of other hexahydroxyflavone.

If this simplification is a result of comparison with any database collecting the existing findings on phytochemistry of M. zapota leaves (literature) - it must be reported.

If the compounds were reported in the literature, each time, analyze if the former proof is reliable enough to be repeated without comment?

e.g. 2

How was epicatechin identified among its four possible stereoisomers? Similarly, there are some salicylic acid isomers, thus why "p-" identified. Similarly, how was assured the position of substitution of quinic acid with gallic, p-coumaric, or caffeic acid?

It is recommended to clear to the readers that a single peak in the direct-injection-MS spectrum can be not resulting from only one compound, e.g., a flavonoid. APCI and ESI profiling in direct-injection-MS can be tricky because numerous isobaric ions can be recorded simultaneously.

= doubts on the presentation =

1

Comparing figures 2a vs. 2c, there is no difference between myricetin and laricitrin.

If stereo-operators (up/down bonds) are used for quinic acid, why not for rhamnose?

2

Collision energies should be reported for each MS/MS. Optimally – as an additional column in the "results" table. A strategy of selection of collision energies should be explained.

Instead of reporting "tentative" compounds in a table, explain the possibilities as well as pros and cons of MS direct injection method.

3

What was the reason to use ethanol extract? Justify. Introduce this information in an appropriate place.

4

All tables and figures must be self-reproducible. Thus, non-popular abbreviations should be explained in captions. In Supplementary Data, too.

5

Fig.S7a, S8a, Tables S2, S3

% of the water in ethanol or ethanol in the water?

6

l.315

literatures:) Each must be stated (e.g., in MS-interpretation table)

7

Sources (references) for FRAP and CUPRAC assays are reported. Sources for DPPH, ABTS, and others not. Supply.

8

Titles of axes in charts in fig 3: Do you measure the % of inhibition of DPPH or ABTS activity?

9

Cheap sentences with unfounded generalizations must be avoided in IJMS papers.

e.g. 1, l.205

"More potency" is not acceptable until you do not clearly state that [22] and your procedure and methods were identical.

e.g. 2, l.253-254

This particular M. zapota leaves exhibited higher activity than particular [22] M. zapota fruits in elastase assay. Can other sorts be of other activity?

e.g. 3, l.390

'In this study, we found that combining ESI and APCI techniques with mass spectrometry 390 maximizes the coverage of m/z peak detection rather than using single ionization technique' - is this conclusion substantially new? I mean the strategy, not the particular case of ZLE.

= minor issues =

After finishing work on Table 1, think about how to compose the headings to avoid unwanted line-breakings. Smaller font can be acceptable for IJMS, I hope.

l.257 and elsewhere

If ABTS•+, why DPPH without radical character?

l.261

If so, declare the 1/ purity classes or 2/ activity units per mass unit for all mentioned chemicals.

= examples of language issues =

l.138

This results suggested

l.140

unnecessary point

l.141

"carbohydrates and starches" => "carbohydrates incl. starches"

fig.4d

correct length units to concentration units

l.182

large ... than

l.218 and below

All Latin botanical names should be presented in italics.

l.223-224

Did the extraction inhibite the enzyme?

Reviewer 2 Report

In this manuscript, the authors have studied an extract of Manilkara zapota leaves in terms of metabolite profiling and biological activity. The extract was shown to inhibit the enzymes a-glucosidase and elastase as well as have antioxidant properties. By comparison to the literature, the authors make some preliminary suggestions on which components of the extract are contributing to the biological activity. Overall, the manuscript is clearly written in both the discussion and the experimental seciton.

However, there is no mention of why only negative mode MS data was collected. A combination of positive and negative ion modes would have detected a larger range of metabolites which would be useful.
